# Automatic Speaker Positioning in Meetings Based on YOLO and TDOA

**DOI:** 10.3390/s23146250

**Published:** 2023-07-08

**Authors:** Chen-Chiung Hsieh, Men-Ru Lu, Hsiao-Ting Tseng

**Affiliations:** 1Department of Computer Science and Engineering, Tatung University, Taipei City 104, Taiwan; 2Department of Information Management, National Central University, Taoyuan City 320, Taiwan; httseng@mgt.ncu.edu.tw

**Keywords:** speaker positioning, time difference of arrival, video conference, YOLO

## Abstract

In recent years, many things have been held via video conferences due to the impact of the COVID-19 epidemic around the world. A webcam will be used in conjunction with a computer and the Internet. However, the network camera cannot automatically turn and cannot lock the screen to the speaker. Therefore, this study uses the objection detector YOLO to capture the upper body of all people on the screen and judge whether each person opens or closes their mouth. At the same time, the Time Difference of Arrival (TDOA) is used to detect the angle of the sound source. Finally, the person’s position obtained by YOLO is reversed to the person’s position in the spatial coordinates through the distance between the person and the camera. Then, the spatial coordinates are used to calculate the angle between the person and the camera through inverse trigonometric functions. Finally, the angle obtained by the camera, and the angle of the sound source obtained by the microphone array, are matched for positioning. The experimental results show that the recall rate of positioning through YOLOX-Tiny reached 85.2%, and the recall rate of TDOA alone reached 88%. Integrating YOLOX-Tiny and TDOA for positioning, the recall rate reached 86.7%, the precision rate reached 100%, and the accuracy reached 94.5%. Therefore, the method proposed in this study can locate the speaker, and it has a better effect than using only one source.

## 1. Introduction

Since February 2020, due to the impact of the COVID-19 virus around the world, many companies can choose to work from home to avoid cluster infections. According to the statistical results of the “Labor Life and Employment Status Survey” conducted by Taiwan’s Ministry of Labor in May 2019, nearly 25% of Taiwan’s labor population can work remotely. However, even if you can work remotely, many things still need to be communicated and assigned, especially for multinational companies that need help traveling abroad for business due to the epidemic. Therefore, discussing business or signing a contract directly through an interview is impossible. The most commonly used method at this time is video conferencing [1] for remote meetings.

Video conferencing involves using a computer and an internet connection in conjunction with a webcam. However, a common issue that arises with webcams is their inability to rotate. This becomes problematic when using a wide-angle lens, as the captured image may not adequately capture the speaker, resulting in a small image of the speaker relative to the overall image. As a result, the viewer may become distracted by other people or objects in the room, making it difficult to concentrate on the meeting. To address this issue, it may be necessary to manually adjust the camera angle to focus on the speaker, which can require additional personnel and disrupt the flow of the meeting [2]. Another way is to utilize motor-control logic to offer both high-speed and ultra-slow, jerk-free pan and tilt movements. This enables camera operators to pinpoint any spot with smooth panoramic viewing. In addition, Axis positioning cameras [3] feature radar auto-tracking for pan, tilt, and zoom (PTZ). It provides automatic tracking [4,5] of detected objects [6,7] for visual confirmation without the need for manual control.

Computer hardware has significantly improved today compared with the past, allowing for the publication of many lightweight and powerful neural networks. As a result, these networks can assist in reducing the need for human intervention in various tasks, allowing the workforce to be utilized where it is most suitable and necessary. Object recognition can be achieved through the use of two-stage [8,9,10] or one-stage detection methods [11,12]. For example, R-CNN [8] uses a two-stage method where it first calculates the possibility that each part of an image is an object, and then hands over potential objects to a convolutional neural network for classification. In contrast, YOLO [12] is a popular one-stage detection method that sacrifices some accuracy for faster object detection. While the accuracy of YOLO was initially lower than the two-stage method, its models have since been improved, and it now surpasses the accuracy of the two-stage method. Consequently, YOLO has become the most widely used neural network for object detection today.

The cost of the current product [2,3,13,14] that offers comparable functions or capabilities is prohibitively expensive for smaller companies. In addition, there are always some inconveniences when using webcams for video conferencing. Therefore, by utilizing increasingly sophisticated neural network technology and high-performance computers, we can effectively enhance work efficiency and reduce video conferencing costs for companies. Some products [3,13,14] have built-in functions that feature remote software control of their pan, tilt, and zoom as well as allowing for hand gestures for control and auto-tracking using subject recognition. However, it may fail when there is more than one person in the image. Additionally, the speaker positioning could not be achieved accurately due to a lack of microphones.

In this study, YOLO is utilized to detect all individuals in a meeting and analyze their current states, such as whether they are speaking with an open mouth, closed mouth, or wearing a mask. Simultaneously, sound source tracking technology is employed to extract the direction of the sound source in the venue. By cross-analyzing each person’s state obtained from the video with the direction of the sound source acquired from the microphone, the individual facing the sound source with an open mouth is recognized as the current speaker. It is crucial to combine both audio and visual inputs for accurate judgments. Relying solely on video may result in a misinterpretation, such as mistakenly identifying a non-speaker as a speaker. Incorporating an audio input significantly minimizes such occurrences.

The novelty of this study Is in the integration of images and sounds to identify the speaker. The camera captures images on the XY plane of the 3D space, while the sound angles are on the XZ plane, as depicted in Figure 1. Since the two types of information are from different planes, they need to be aligned and positioned correctly. To achieve this, YOLO is used to distinguish each person’s mouth, and the speaker’s upper body is identified in the video image during a meeting, resulting in an overall recall rate of over 80%. This approach prevents closed-mouth situations from being erroneously identified as open-mouthed. Additionally, the study aims to quickly and accurately determine the sound source direction through a microphone array with a recall rate of at least 80%. Finally, by combining the information from both the image and sound, the speaker’s location can be confirmed with a higher overall recall rate than using only images for identification. In this study, we have only a horizontal microphone array that corresponds to the XZ plane, so the YZ plane provides no information. However, the YZ plane is useful if a vertical microphone array is deployed.

This study is structured into five sections. The Section 1 presents the background, motivation, research objectives, and innovations of the research. The Section 2 is dedicated to the literature review, which is divided into three parts. The first part reviews the current state of object detection, the second part focuses on various methods of sound source localization, and the third part describes the latest advancements in speaker identification. The Section 3 provides a detailed description of the system architecture, including the real-time YOLO model for face and mouth detection, the depth camera, and the microphone array. The number of classes for the YOLO model and the datasets collected for each category are also described. In the Section 4, the experimental results and analysis are presented in detail, including a comparison of speaker identification results using different neural network models. The Section 5 concludes the study and outlines potential future work.

## 2. Related Research

### 2.1. Object Positioning by Images

To find objects in an image, there are two main approaches for this goal. The first is the two-stage approach [8,9,10], which is the early object detection approach. The accuracy of the results obtained in this way is usually higher, but the speed is slower. The other is the one-stage approach, which is the more recently proposed approach to object detection. It is characterized by only one neural network that can directly detect the location and class of objects. The accuracy of this method is usually lower, but the speed is much faster than that obtained by the first method. In the following, we describe some recently developed deep learning models based on YOLO.

#### 2.1.1. YOLO

YOLO [12], which stands for You Only Look Once, is a neural network that only needs to perform one CNN operation on an image to obtain the object’s class and location. The original author of YOLO has published three versions of YOLO, with each version improving the overall detection performance. YOLOv1 [12] proposed a new object detection method that differs from the traditional sliding window and region proposal approaches. YOLOv1 performs a convolution on the entire image during both training and inference processes, resulting in a background error detection rate only half that of Fast R-CNN [9]. YOLOv1 divides the input image into S × S grids, with each grid predicting B boundary boxes and confidence scores. The predicted boundary boxes and confidence scores are then adjusted based on ground truth data to obtain an S × S × (5 × B + C) tensor, which contains the object’s location and class in the image. The redundant boundary boxes are removed using non-maximum suppression (NMS) [8] to obtain the final result.

The main advantages of YOLOv1 over R-CNN are that it is easy to train, fast, and has a low background error detection rate. However, its disadvantages are that it has a lower detection accuracy for small objects and larger localization errors. To improve the precision and recall of object detection, the original authors of YOLO proposed YOLOv2 [15], which introduced the concept of an anchor box from Faster R-CNN [10]. This modification enabled the model to predict the boundary box more accurately and easily, resulting in significant improvements in speed and accuracy. YOLOv3 [16] is a further optimization of the YOLOv2 model, which includes the use of the ResNet [17] architecture to address gradient problems, deepening the network from around 20 layers to 53 layers, and using multiscale feature maps for detecting objects of different sizes, thereby enhancing the prediction capability for smaller objects.

Several years after the original author proposed YOLOv3, the Academia Sinica in Taiwan and foreign scholars jointly published a new version of YOLO, named YOLOv4 [18]. Based on the previous work of YOLOv3, YOLOv4 has been further improved to achieve a faster execution speed and higher accuracy.

#### 2.1.2. YOLOX

YOLOX [19] is an algorithm that improves upon the YOLO algorithm proposed by a Chinese technology company in 2021. The main contribution is the optimization of certain parts of YOLOv3, including the decoupled head, strong augmentation, anchor-free, multi positive, and SimOTA (similarity-based object-targeted augmentation) parts. The primary modification to the decoupled head involves separating the classification and regression tasks of YOLOv3. Although separating the tasks increases the computational complexity, the dimensionality of feature maps can be reduced first, allowing for a significant increase in the AP and convergence speed with only a minimal increase in the parameter quantity.

YOLOX improves upon the dataset part through strong augmentation, which includes two methods: Mosaic and Mix-up. Mosaic randomly scales and crops four images and arranges them in a random order to create a new training image. Mix-up blends images with different opacities. Strong augmentation turns off data augmentation in the last 15 epochs of training because the methods used can increase the diversity of images and the number of samples in the dataset, but may result in many false-positive cases due to the large amount of cropping and synthesis.

Anchor-free was used in YOLOX because using anchors requires k-means analysis on the ground truth to determine the best anchor points. This can make the model lack generalization, and this mechanism can increase the complexity of the detection head and the number of generated results. The adjustment was made by reducing the number of predictions per position from three to one and directly predicting the offset from the upper-left corner of the grid and the length and width values of the bounding box. Additionally, the center of each object was set as a positive sample, and the objects were assigned to the correct FPN according to their size. This improvement not only reduced the number of predicted results from 2,142,000 to 714,000, greatly reducing the number of parameters of the detector and the computation cost, but also made it faster and obtained better performance.

#### 2.1.3. YOLOR

YOLOR [20] and YOLOX were proposed almost simultaneously, but they are very different. YOLOX is similar to the YOLO series of object detection neural networks presented earlier, which was built on previous foundations and altered some of the architecture, increasing the speed or accuracy or making the predicted bounding box more accurate. However, YOLOR goes one step further by making it possible for the model to learn the characteristics of a particular object and all the attributes of the input data. When the model needs to detect non-current learning targets, it only needs to extract the features it needs to use from the integrated features, and then it can detect these new targets without the need to use transfer learning to retrain the new target. According to official figures released by YOLOR, it is 315 percent faster than Scaled-YOLOv4 with the same accuracy, and it is the top one in the contest on the COCO dataset.

YOLOR is divided into three parts: explicit knowledge, implicit knowledge, and knowledge modeling. First of all, explicit knowledge is a feature related to the input value for YOLOR, and implicit knowledge is a feature that is not associated with the input value. The knowledge modeling mainly includes using sparse methods to pre-define and model the learning dictionary or use the memory network to combine various features to form an integrated feature model and make it such that a feature model can be added or changed dynamically. YOLOR can be learned through multi-task learning and has a unified network architecture. Therefore, implicit knowledge can be applied to different places separately, allowing YOLOR to choose different implicit knowledge weights for prediction based on different tasks.

### 2.2. Sound Source Localization Method

There are several methods for sound source localization, such as Beamforming, Super-resolution spatial spectrum estimation, and TDOA (Time Difference of Arrival). This study analyzed and compared the differences between Beamforming and TDOA, and ultimately chose the TDOA method for sound source localization.

#### 2.2.1. Beamforming

The basic concept of Beamforming [21] is to form a beam by a weighted summation of signals collected by each microphone. By searching the possible positions of the sound source to guide the beam, and modifying the weights to maximize the power of the sound output, the sound source location may be determined. This method has strong adaptability to the environment and can achieve better noise resistance, but it is more suitable for large microphone arrays. The cost of this kind of microphone array is higher than that of a general microphone array, and the space required for such a microphone array is also larger. Therefore, using this type of microphone array [22] for simple sound source localization will only increase the physical size of the product. In addition, this localization method is more suitable for long-distance sound source localization.

#### 2.2.2. TDOA

TDOA [22], short for Time Difference of Arrival, calculates the direction by measuring the time difference between signals. This method is not only suitable for sound source localization, but also for signal source localization. For sound source localization, this method first analyzes the sound and extracts certain features. As shown in Figure 2, it compares the time difference *δ* of the same feature in different audio tracks obtained by the microphone array (m1~m3). The angle *θ* can be calculated using this time difference as in Equation (1), the known speed of sound *C*, and the distance *d* between the microphones of the array. Since it involves sound features for pattern matching, the signal sampling frequency *f_s_* is also needed. At least two microphones are required for detecting the angle *θ* of the sound source using TDOA, and the more microphones used, the more precise the angle obtained. Furthermore, the angle obtained using TDOA becomes more complex with different microphone arrangements.
(1)$$\theta =\sin^{-1}\left[\frac{\delta \times C}{f_{s}\times d}\right]$$

### 2.3. Speaker Localization Method

Obtaining the location of individuals in a real-world space from images captured by a camera is a challenging task. It is unlikely that we will be able to achieve an accurate localization by relying solely on one method; thus, many papers are currently researching various ways to locate individuals in different ways. One of the most popular and widely used research methods today involves using smartphones for localization. Since most people own smartphones, which typically contain various kinds of hardware, such as a network, a GPS, accelerometers, and magnetometers, it is easy to write programs using SDKs such as Android Studio. Therefore, development can be simpler and more convenient.

One of the approaches is to deploy an RFID reader embedded in a smartphone. Ma et al. [23] proposed a method that requires individuals to carry such smartphones with RFID readers and to place RFID tags throughout the venue. The reader will then match the pre-set location information of the tags and obtain information such as the distance between the reader and each tag. By calculating the distance, the position of the individual can be determined. According to the paper, as long as RFID tags are effectively placed in the venue and the reader can detect multiple tags at the same time, the system can accurately locate the position of individuals. This method is suitable for different scenarios, has good scalability, and can provide high accuracy. However, the prerequisite for this method is that the speaker needs to carry an RFID reader. Forgetting to bring it would affect the entire meeting’s progress.

Antsfeld et al. [24] utilized various hardware components of a smartphone, such as inertial sensors, barometers, and Wi-Fi network sensors, for data collection. The collected data were then used to predict the movement trajectory of individuals using a deep learning model. After obtaining the predicted trajectory, a Kalman filter was applied to correct for drift using Wi-Fi signals, enabling the prediction of the absolute position of individuals with each Wi-Fi scan. Finally, a map-less projection method was used to adjust the results of the Kalman filter and project the predicted results onto walkable locations. The method is similar to the previous one and also requires users to carry a mobile phone. In addition, everyone may be sitting throughout the meeting, so if a method that relies on detecting changes in body movements or postures is used to locate individuals, it will often be unable to correctly locate speakers in most cases.

Subramanian et al. [25] proposed a new supervised learning approach using deep neural networks to distinguish different speaker sound source directions in the audio obtained from microphone arrays. They believe that multi-speaker dialogue analysis is an important and challenging kind of technology. Compared with existing deep learning methods that can only achieve a few frames per second, their proposed method is so fast that it can discern angles word by word. They also added automatic speech recognition as an additional function, which resulted in a word error rate of only 6.3% in tests where two people spoke simultaneously. The advantage of this method is that it can detect the positions of multiple speakers in the field simultaneously. However, because only the sound is used to distinguish the angle of the sound source, it is still not possible to obtain the exact position of the speaker in the video.

The target field of this study is a relatively quiet area, a meeting room, so there is less noise and fewer situations where multiple people speak at the same time. Our goal is only to locate the current speaker, so recognizing the sound of every person would be a waste of computational resources.

## 3. Proposed Methodology

The aim of this study is to propose a simple method that can use only a camera and a microphone array to capture the current status of all individuals in a meeting room, and accurately and quickly extract the speaker. In this section, we will explain the architecture and process flow of the entire system. We will also discuss the importance of data collection in training neural networks, how to choose the appropriate neural network, and modifications required for real-time recognition. Finally, we will explain how to integrate the results of YOLO object detection and sound source localization to improve the accuracy of recognition and reduce the probability of a misjudgment.

The relevant research in the previous section shows that most of the existing speaker positioning methods require the speaker to carry an additional item. Moreover, these methods all have a problem. That is, if the speaker forgets to carry it, it will seriously affect the progress of the entire meeting. Another problem is that many practices only do positioning. For our usage scenario, there is no way to capture the image of the speaker well. The most significant improvement of this study is that we do not need to use additional objects when positioning. We only need a camera and a microphone to achieve a high-precision positioning effect, and we can directly capture the image of the positioned speaker as an output.

### 3.1. System Architecture

Figure 3 shows the system activity diagram of this study. Firstly, the video camera is used to capture images of the meeting room, and the microphone array is used to record the sound in the meeting room. These two devices capture the conference room situation (as an image and sound) at the same time. Next, the positions of the persons in the scene are obtained through the image, and the direction of the sound source is obtained through the microphone array. Finally, the sound source direction filters the persons obtained from the image. The retained person is the identification result for the output and repeats the above operations for the next time frame.

Along the image track is person detection by YOLO, which requires image preprocessing, such as resizing, and post-processing after several candidates are detected. Along the sound data processing track, feature extraction for TDOA-based sound source positioning is the main task. For these two independent tracks, we could implement them as threads for concurrent processing. After collecting (joining) enough information of the same frame from both tracks (threads), we can do person filtering to find the current speaker.

#### 3.1.1. Hardware Installation for Sound Source Positioning

This research will first align the center of the range that the camera can capture with the 90° position of the microphone array device as shown in Figure 4. Then, the shooting range of the camera is used to set the upper and lower limits of the microphone value. For example, for a wide-angle camera with a maximum shooting angle of 120°, the upper and lower limits of the microphone are set to 150° and 30°, respectively. After setting the upper and lower limits of the microphone, the (x, y) coordinates on the photo will be converted into XYZ coordinates in the space through the depth camera and the API provided. Finally, the arctangent in the trigonometric function is used to convert the coordinates into angles so that the result of object detection can be matched with the angle of sound source localization through a calculation.

#### 3.1.2. Person Positioning

The positioning of the speaker is mainly based on the image obtained by the camera, and the process is as follows. First, obtain the current status image of the meeting room through the camera, and then preprocess the image. Before performing the prediction of the YOLO neural network, the input image must be scaled to the size set during YOLO training. After the scaling is completed, it will usually be smaller than the original set size, because the images captured by general cameras are all in a ratio of 4:3 or 16:9. Therefore, if the length and width are set to be the same, there will be blank areas, and these blank areas need to be filled with gray.

The preprocessed pictures are handed over to the trained YOLO neural network for prediction. The predicted result includes information such as the position of the closest Grid, the offset from the upper-left corner of the Grid, the size of the prediction frame, and the confidence value. Then, the position and size of the prediction box are calculated to the position of the original image through some coordinate transformations. We use the NMS method to remove duplicate predicted boxes. Finally, the predicted results and the direction of the sound source are screened again. If there is no speaker in the screened results, the next prediction is directly carried out.

#### 3.1.3. Sound Source Positioning

Sound source localization mainly uses the microphone array to record the sound, and then calculates the angle of the sound source through the TDOA algorithm. The TDOA calculation process first converts the sound into the frequency domain through the Fast Fourier Transform (FFT). Then, the correlation between the samples is calculated through the frequency domain. Then, we calculate the position with the greatest correlation among the several samples before and after each channel, and we obtain the time difference when the sound arrives at the microphone.

TDOA Algorithm.

Do FFT for each channel and obtain fft_real and fft_img;For each channel *i*For each sample point *j*Do FFT cross-correlation for channel *i*;Do inverse FFT for channel *i*;Rearrange IFFT and make half of the points the center;Select the max correlation *max_j* within a margin interval;TDOA[*i*] = *max_j*;

After obtaining the microphone time difference, combined with the speed of the sound and the distance between the microphones, the angle between the sound source and the microphone can be calculated by using a simple trigonometric function relationship as shown in Equation (1). The accuracy is mainly affected by two factors: the sound sampling frequency *f_s_* and the distance *d* between microphones. Usually, the higher the values of these parameters, the higher the accuracy we obtain. After obtaining the angle of the sound source, we can filter the prediction results of YOLO.

#### 3.1.4. Filtering False Positives

Although the accuracy of YOLO’s recognition results is not bad, there will still be false-positive recognition results for the mouth opening category. This research hopes not only to use the images obtained by the camera as the only reference basis but also to deploy YOLO for preliminary screening. The results extracted by YOLO are then filtered again through sound source localization. The probability of false positives is reduced, additional training is not required for the speaker, and the speaker does not need to carry certain devices for additional judgments.

As for filtering, the angle calculation results of TDOA will be cross-validated with the prediction results of YOLO. Verification is done by considering the person whose mouth is open and who is in the same direction as the sound source to be the speaker. If there is a person who is wearing a mask or has a closed mouth with a low confidence value in the same direction as the sound source, and there are no other people who open their mouths nearby, this person will also be regarded as a speaker. Finally, we capture the half-length image of the speaker prediction box as the output image as shown in Figure 5.

### 3.2. Neural Network Training

This section introduces two critical parts in training neural networks, namely data collection and the selection of neural network models. Data collection is the key to improving the accuracy of training results. If the data collection is comprehensive enough, it will lead to sufficient training accuracy. If the similarity of the collected data is too high, it may lead to overfitting of the training results. There are many kinds of neural networks for object detection. Excellent results have been achieved so far, regardless of whether a two-stage object detection network or a one-stage network was used. Choosing a neural network suitable for our use situation is also a problem.

#### 3.2.1. Data Collection

For our usage environment, the data collection involved searching for videos such as public hearings, speeches, and news clips through the Internet. The photos of these multi-person meetings as shown in Figure 6a allow the neural network to recognize multiple people’s bodies and mouth shapes simultaneously. Video can be converted to pictures to increase the amount of data quickly. In order to avoid the dataset being too similar, a frame was captured into a picture every few seconds as shown in Figure 6b. There was also a multivariate background added to avoid overfitting. In order to enable use in the international community, in addition to using local photos for training, many videos and photos of foreign speeches were also added as shown in Figure 6c to increase the diversity of the dataset. In this study, 1899 photos were initially collected through the above method, and the number of photos was increased later depending on the quality of the training and test results.

After the initial data collection, YOLOv4-Tiny was first used for training. The main reason is that its training speed is faster and has a certain degree of accuracy. Therefore, it is very suitable for rapid training to verify whether there is a problem with the dataset. After the initial training, we found several problems, including the fact that a closed mouth with a frontal face will be recognized as an open mouth (Figure 7a, left), the body type cannot be correctly recognized when the hands are raised (Figure 7a, middle), people who are far away will not be recognized correctly (Figure 7a, right), and the fisheye effect of the wide-angle image will not be detected. Except for the problem of distance, which could not be shot due to site restrictions, we solved the other problems by shooting the problem images and adding them to the dataset. For the distance problem, we used the strong augmentation of YOLOX to expand the existing picture (Figure 7b) to achieve the same effect as the long-distance picture. In the end, the total number of pictures we collected for training and pictures added or expanded to solve the errors found in training reached 4673.

In the end, among the 4673 photos, there were 2864 mouth-opening labels, 1437 shut-up labels, 1390 mask labels, and 5387 upper-body labels. The data distribution is shown in Figure 8. There are many categories of mouth opening because we focused on detecting the state of mouth opening, and there is no particular mouth shape when closing the mouth. As for the appearance of the masks, they were all similar, with only differences in design and color, so they were easier to identify. In addition, the label of the upper-body category is several times the size of other categories. The reason for this is that, regardless of the category of opening the mouth, shutting the mouth, or wearing a mask, as long as there are photos of the half-body picture, the label of the upper-body category needs to be marked. Therefore, the label of the upper-body category is almost equivalent to summing the number of labels for the other three categories. In this study, the data distribution was not balanced and data augmentation is often used to increase the dataset’s size. Hopefully, the gaps among the categories are not too large and resizing, rotating, and flipping the images can close them.

#### 3.2.2. Neural Network Model Selection

As mentioned in Section 2, many neural network models exist for object detection. Although the accuracy of the two-stage object detection network is better, the execution speed of the one-stage method will be much slower due to the problem of the overall structure. This study aims to achieve instant identification, so it is less applicable than one-stage object detection. Except for older versions such as YOLOv1~YOLOv3, YOLOv4 and newer versions have experienced major improvements and were considered to be used in our system. There is also a version of YOLOv5 to which some improvements have been made to the architecture. Although no papers have been published, the program has been released, and a website has been provided that contains an explanation. Some of the changes also have been used in YOLOv4, and the adjusted test results were even better than those of YOLOv5.

Based on the data provided by the YOLO official neural network models for object detection [10] and comparing the accuracy of the neural network models at the same processing speed, YOLOR has the highest AP. However, these training and testing results are based on the COCO dataset. Different results will still depend on different environments and detection targets. Therefore, we used several one-stage object detection neural networks after YOLOv4 to train and test our dataset. We tested six different neural networks, including YOLOv4, YOLOv4-tiny, Scaled YOLOv4, YOLOv5, YOLOX-Tiny, and YOLOR. The actual test results and our final analysis and selection are described in detail in Section 4.

## 4. Experimental Results and Analysis

This section introduces the independent functions of this research, the test methods, the test environment, and the test results of YOLO and TDOA. Moreover, we discuss the benefits of the method proposed in this study. First, we explain the test results of the mouth state and the upper-body image through YOLO and then explain the test results and accuracy of locating the source direction of the sound through TDOA. Finally, the results of co-location combined with the above two parts are explained.

### 4.1. YOLO Series Models Testing

This section describes the various experiments and experimental data of each experiment conducted during the YOLO test. Moreover, we explain the object detection neural network selected through the experimental results.

#### 4.1.1. YOLO Series Models Test Environment

In this study, two different cameras were used when testing YOLO. One was a camera with a resolution of 1280 × 720. This camera was a wide-angle camera with a horizontal viewing angle of 70 degrees as shown in Figure 9, and the picture presents a fisheye effect. The other was a camera with a resolution of 1920 × 1080. The horizontal viewing angle also had a viewing range of 70 degrees, and the picture presents a corrected fisheye effect. The above two cameras can be used to check whether the trained neural network will have a poor recognition rate or even be completely ineffective due to the difference in the camera resolution or lens. The following section describes how detection was tested using YOLO and presents the results obtained from various YOLO tests.

#### 4.1.2. YOLO Series Models Test Method

This study performed angle and distance tests on people, respectively. The test subjects were in front of the camera, the left front of the camera, and the right front of the camera, respectively. Furthermore, we let the subject’s head tilt from top to bottom, divided into three angles (downward, upward, and head-up) to the camera, in order to simulate the differences in elevation angles of the images obtained by the camera due to the different heights of each person. The distance was from one meter to four meters from the camera, and four different distances were tested. Each distance was tested in the three directions and different elevation angles mentioned above. The above tests were all tests that need to be conducted for one mouth state, so three different mouth states in this study were combined into a total of 108 different images as shown in Figure 10a–d. This way, the various detection accuracies obtained under different positions and angles during the meeting can be simulated.

#### 4.1.3. YOLO Series Models Test Results

This study tested six neural networks using the dataset we produced. The performance results of the training and the data during the test are explained according to different situations. “TP” means that a specific type is detected, and there is indeed the object in its position, “TN” means that no object is detected and there is no such object in its position, “FP” means that a specific type is detected, but there is no such object in its position, and “FN” indicates that the object exists in a specific position but has not been detected correctly. Performance indicators such as mAP and precision can be calculated using these data to make the data more valuable. Among them, mAP represents the average AP of all categories. AP is obtained by calculating precision and recall through different confidence values and the area covered by the PR curve. Table 1 shows the results of various neural networks tested with the test dataset. The test results show that YOLOv4, Scaled YOLOv4, and YOLOR-P6 are all worse than the other neural networks. Therefore, this study did not include these types of neural networks in the scope of testing and consideration when using the camera for actual operation and testing.

In this study, the training results were used for an initial screening. Three more suitable neural networks were selected, namely YOLOv4-tiny, YOLOv5, and YOLOX-tiny. Then, we tested these three neural networks using the test environment and test methods proposed in the previous sections. We recorded the recall rate of each neural network, and the results are shown in Table 2, Table 3, Table 4, Table 5, Table 6 and Table 7. By analyzing the test results of each neural network, we can verify which neural network has a better effect in actual use.

From the above test results, we can draw the following conclusions. If we use a camera with a low resolution and little picture distortion, the neural network with the best performance among the three neural networks is YOLOX-Tiny. While the other two neural networks were less able to detect objects correctly, YOLOX-Tiny always worked. If a camera with a higher resolution and no image distortion is used for testing, the neural network with the best test results is also YOLOX-Tiny. Based on the above results, the recall rate of YOLOX-Tiny is significantly higher than the recall rate of the other two neural networks. Therefore, YOLOX-Tiny should be more adaptable to various shooting environments, so the neural network chosen for integration with TDOA in this study was YOLOX-Tiny.

### 4.2. TDOA Testing

This section will explain the test environment, such as the microphones and the arrangement used in the TDOA test, and explain the test method. Finally, the test results and the conclusions drawn from the test results will be explained.

#### 4.2.1. TDOA Test Environment

The TDOA in this study used a microphone array with a total of four microphones, and the arrangement of the microphones is in a straight line, as shown in Figure 11a. Therefore, the angle range that this microphone array can detect is only a horizontal angle of 180 degrees, and it is impossible to distinguish whether the sound source is from the front or the rear. However, it will be placed at the front of the meeting room for conference use, so it does not need to consider whether the sound source is in the front or rear. In addition, the distance between each pair of microphones in the microphone array is fixed at 7 cm. Because the TDOA in this study was only used for auxiliary positioning, only a rough angle was needed, and it did not need to be accurate enough to distinguish a difference of one degree. Hence, the microphones used in this study were only the leftmost and rightmost two microphones.

#### 4.2.2. TDOA Test Method

The way to test the accuracy of the TDOA is to place the sound source two meters away from the microphone. The sound source was tested in units of 10 degrees from 30 to 150 degrees (Figure 11b), so in total 13 angles were tested. The reason for discarding the positions of the sound sources at less than 30 degrees and greater than 150 degrees is that the camera can only obtain the picture at a manageable angle. Therefore, this study focused on the angles that the camera can cover.

The angle obtained by the TDOA of this study only needs to have a roughly accurate angle. Therefore, this study divided the available angles into 19 angles and divided the angles closer to the center of the picture into smaller blocks. Each different angle was tested ten times. When the difference between the angle of the test result and the angle of the actual sound source was within five degrees, it was recorded as the correct positioning.

#### 4.2.3. TDOA Test Results

The test results of the TDOA are shown in Table 8. The green part is the predicted angle within five degrees of the actual angle, and the overall recall rate is 88%. From the test results, this study used the TDOA to extract the angle of the sound source, and its accuracy reached the level that can assist YOLO in positioning.

### 4.3. Integrated YOLO and TDOA Positioning Test

This section describes the environment and equipment used in the integrated positioning of the two positioning methods used in this study. We explain how the positioning accuracy can be improved by integration, and what test results can expected from this research. Finally, the integration test results are explained, and the pre- and post-integration tests are analyzed and compared.

#### 4.3.1. Integrated Positioning Test Environment

To integrate the positioning results of YOLO and TDOA, we need to obtain the position and angle of the object detected by YOLO in space. To obtain the position and angle of an object in space through images, it is first necessary to obtain the distance between the object and the camera and then inversely deduce it through the camera’s visible range and the object’s position in the image. The result of the inverse inference is the coordinates of the object’s position in space relative to the camera as shown in Figure 12a. After obtaining the object’s coordinates in space, the XZ (horizontal) coordinates can be used to calculate the arctangent function, which is the object’s angle relative to the camera.

This study obtained the distance *d* between the object and the camera using an Intel RealSense D435 (Figure 12b), a camera that combines infrared and visible light. Through this camera, we can use YOLO to make predictions in color images. Then, the person’s position *(x, y)* in the picture is obtained through the result of the YOLO prediction, and the coordinates and angle *θ* of the person in the space can be calculated by Equation (2).
(2)$$\theta =\tan^{-1}\frac{x}{d}$$

The camera’s height was fixed at one meter above the ground during the experiment. Furthermore, we tested the conditions of sufficient and poor light in two different spaces (Figure 13). Space A has a brightness of 765 lux in good lighting conditions and 517 lux in poor lighting conditions. Space B has a brightness of 383 lux in the case of sufficient light and a brightness of 220 lux in the case of poor lighting conditions. The above several scenarios prove that the recognition results in different spaces and lighting conditions have a certain degree of reliability.

#### 4.3.2. Integrated Positioning Test Method

The complete test video was stored in photos during the integrated positioning test, and two independent experiments were carried out. The first experiment only used YOLO for identification, and the person identified as having an open mouth was regarded as the speaker. The other experiment used YOLO for identification and matched the position of the person identified as having an open mouth to the source of the sound. The matching method considers the result of the difference between the angle obtained through the video and the angle obtained through the sound within 10 degrees as a successful match. The result of a successful pairing is considered to be the speaker. Alternatively, a person identified as having their mouth closed but with a low confidence value and in the same direction as the sound source is also considered a speaker. In addition, if YOLO does not detect a person with an open mouth, but the result of the sound source location is in the same direction as a person wearing a mask, it will also be regarded as a speaker to prevent the speaker wearing a mask from being incorrectly identified. The above two experiments were used to calculate the accuracy and other indicators before and after integration. We expected to prove that the integration of YOLO and TDOA in this study can achieve better results through the various indicators before and after the integration.

Statistics are presented in a confusion matrix. TP is the result of the current speaker and is detected as a speaker. FP results from a non-speaker; however, it is identified as a speaker. FN results from the current speaker not being detected as a speaker, and TN is a non-speaker and indeed not detected as a speaker result.

The test was divided into two parts. First, there was only one person in the image, and the person walked within the visible range in order to test the tracking effect of this method. Three angles were tested in the single-person test method: 60 degrees, 90 degrees, and 120 degrees. Each angle was used to test whether a mask was worn, and each state tested for about 100 frames. In the other part, there were five persons in the frame simultaneously. The subjects were asked to speak in sequence from left to right during the test. Each person was tested on whether they wore a mask for about 100 frames. In this way, the success rate and error rate of this identification method could be tested in the situation of multiple people.

#### 4.3.3. Integrated Positioning Test—Single Person

Table 9, Table 10, Table 11 and Table 12 present the results under various test environments. The first two columns of each test are the test data using only YOLO for positioning, and the last two columns are the positioning results of using the direction of the sound source to assist YOLO. All data were divided into test results that include masks and test results that do not. This is because if only YOLO is used for positioning, the person who wears a mask but is a speaker has no other basis on which to be judged as a speaker. Therefore, he/she will not be detected so these cases will be compared separately. Through various statistical results, we can obtain various indicators for judging the quality of the proposed method. The data show that the integrated approach to speaker positioning is better than without the integration.

#### 4.3.4. Integrated Positioning Test—Multiple Persons

This section is consistent with the test method of the previous section. It tested only the results of YOLO for positioning and used the sound source direction to assist YOLO for positioning. Through multi-person testing, we obtained the same results as in single-person testing. Various statistical metrics are shown in Table 13, Table 14, Table 15 and Table 16 to evaluate the proposed method better.

#### 4.3.5. Integrated Positioning Test—Analysis

The first two sections illustrate the experimental results of this study under different spaces, lights, and numbers of people. The integrated precision and accuracy are both very high, thanks to the positioning of the sound source. This is because, after the YOLO positioning, if the mouth of the person in the picture is misjudged as an open mouth, or the person in the picture is yawning and has other abnormal mouth-opening conditions, it will be misidentified. However, after integrating TDOA, it is necessary to determine whether there is any sound so that false positives can be significantly reduced.

The results of experiments with an insufficient amount of light show that some TPs that use sound-source-assisted YOLO for positioning are higher than those that only use YOLO. The reason for this is that, in the case of insufficient light, it is more likely that the opening of the mouth will be recognized as the closing of the mouth. The confidence value of these YOLOs for these results is usually low, so after using the sound-source-assisted YOLO, these results are identified as shutting-up. However, with low confidence, values can be changed for speakers.

The experimental results show that when the speaker is wearing a mask, the recognition effect of the mask is better. Therefore, the results of localization using sound-source-assisted YOLO are very good. Furthermore, when the speaker is not wearing a mask, the characteristics of his mouth will be affected by the light or mouth shape, resulting in poor recognition results. As a result, some mouths will be misjudged to be closed when they are open.

## 5. Conclusions

This study integrated YOLO and TDOA for speaker positioning. We used the sound source angle analyzed by TDOA to match the speaker positioned by YOLO. According to the experimental test results, the recall rate reached 86.7%, the precision rate reached 100%, and the accuracy reached 94.5%. When only YOLO was used to obtain the results of opening the mouth to locate the speaker, the test results were 40.5% for recall, 54.8% for precision, and 61.4% for accuracy. The above data prove that using the sound source angle to strengthen the matching of YOLO’s speaker positioning can significantly reduce the probability of being misjudged as a speaker. Moreover, the entire positioning system can still be used when the speaker is not wearing a mask.

The method proposed in this study can indeed locate the speaker and, compared with positioning using RFID or the inertial sensor of a mobile phone, this can be done without requiring the speaker to carry additional items. In the past, traditional methods usually only did positioning. Our method not only locates the speaker’s position but also captures and outputs the speaker’s image in real time. In addition, the only equipment needed for this method is a camera and a microphone array. Even with a better microphone array and depth camera, the hardware costs only about USD 200. Compared with the existing equipment that can perform similar functions, which costs more than USD 600, the method proposed in this study can effectively reduce the equipment cost required for video conferencing.

This method currently uses YOLOX-Tiny as a neural network for extracting various target objects in images. Although the recognition effect is excellent, it still cannot obtain the state of the mouth or the body’s position when the distance is great or the brightness is low. In the future, we will test other object detection neural networks and increase the amount of data in the dataset in order to improve the recognition effect.

This method uses only the two farthest microphones for sound source angle detection. In the future, we will also use more microphones to improve the recognition accuracy. In addition, the microphone can also be replaced, and the microphone can be replaced with a microphone array that can detect the full range (360 degrees horizontally and 360 degrees vertically). The angles obtained with this microphone array can better match the positions obtained by YOLO. In this way, it will also be possible to distinguish between orientations and people of different heights.

At present, we obtain the coordinates of the person in the prediction frame in space based on the distance between the person and the camera obtained by the depth camera. Then, we use the distance to reverse the person’s position in space. However, the price of the depth camera is relatively high, so if the distance between the person and the camera can be obtained through other methods, the method proposed in this study can use ordinary cameras in the future. We can also estimate the size of the face detected by YOLO or further estimate the distance between the eyes and nose of the face. After estimating the distance between the person and the camera, the processing remains unchanged. However, the camera can be changed to an ordinary camera to reduce the cost significantly.

## Figures and Tables

**Figure 1 sensors-23-06250-f001:**
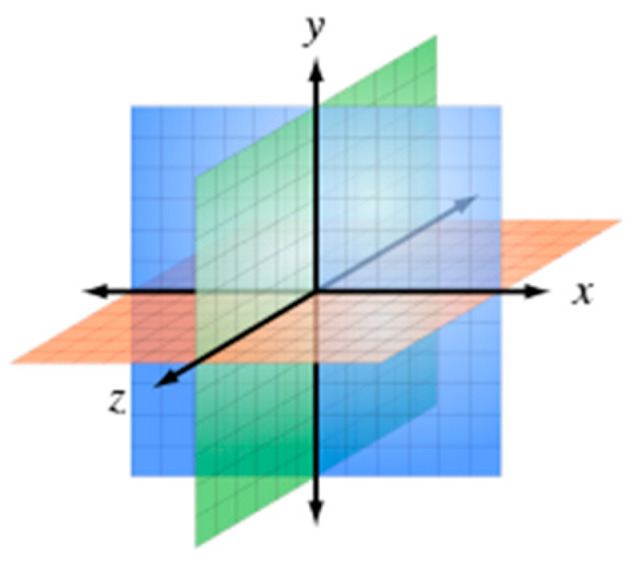
The 3D XYX planes.

**Figure 2 sensors-23-06250-f002:**
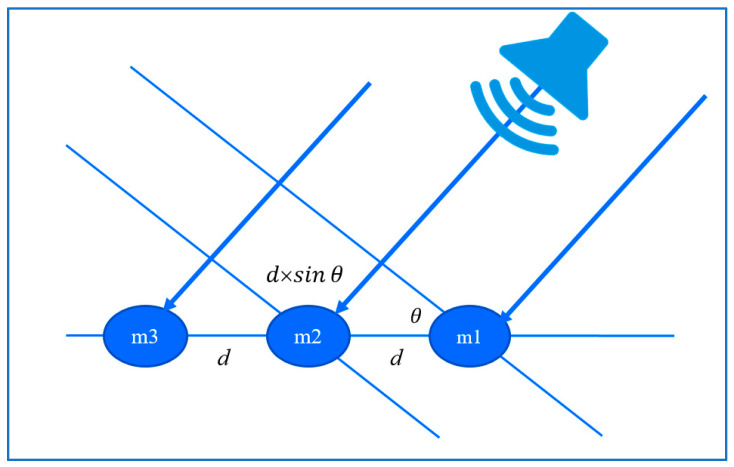
TDOA angle calculation.

**Figure 3 sensors-23-06250-f003:**
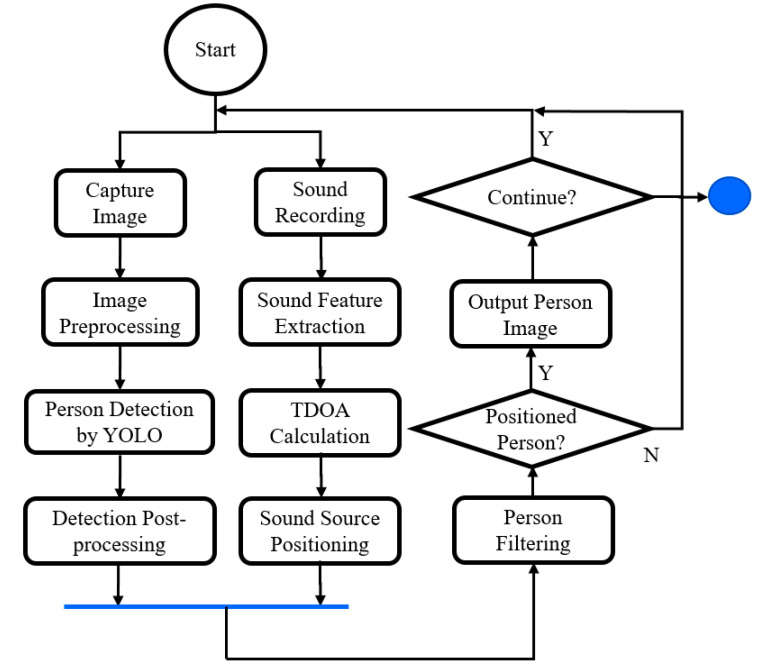
System activity diagram.

**Figure 4 sensors-23-06250-f004:**
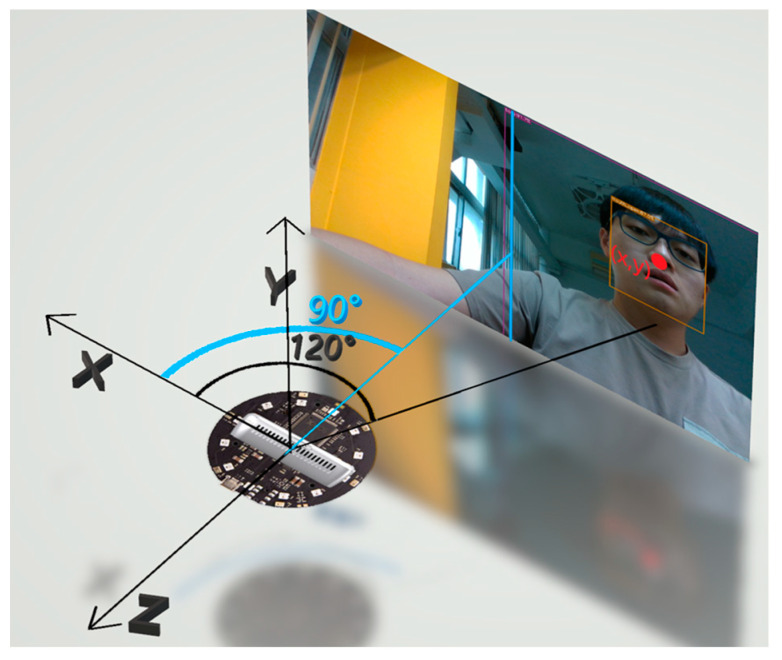
Alignment of the camera and microphone array. The circle-shaped object is the microphone array and the rectangular object is the used camera.

**Figure 5 sensors-23-06250-f005:**
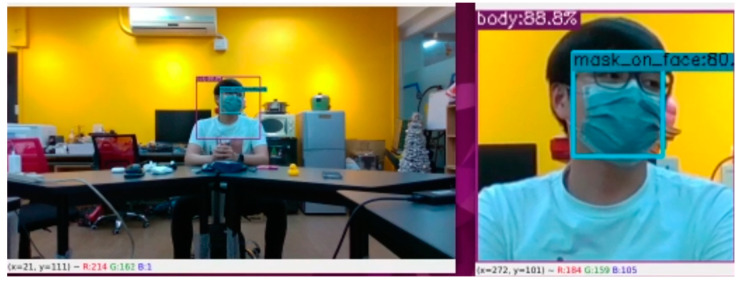
Sound source positioning-assisted filtering after YOLO person detection (**left**) and the output low-confidence masked target person (**right**).

**Figure 6 sensors-23-06250-f006:**
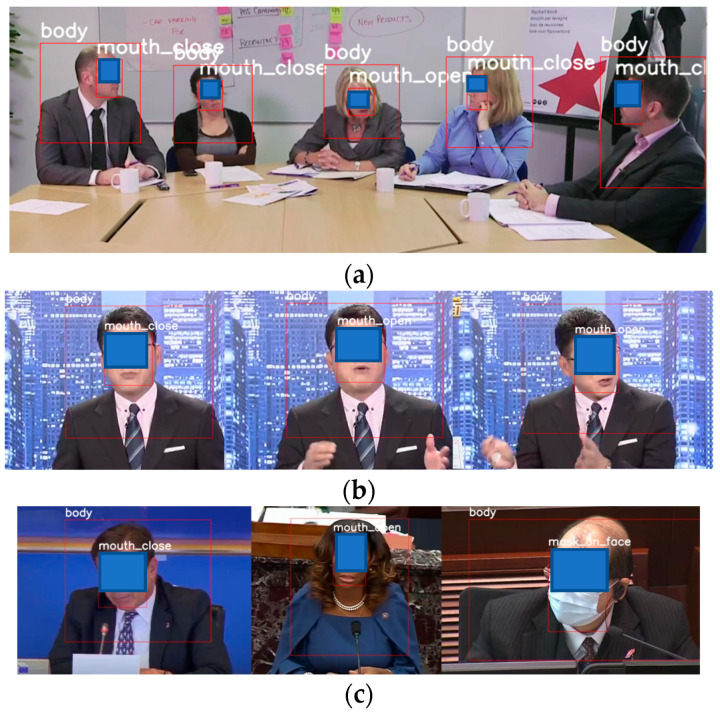
Data Collection. (**a**) News clips. (**b**) Multi-person meeting. (**c**) Multi-ethnic data.

**Figure 7 sensors-23-06250-f007:**
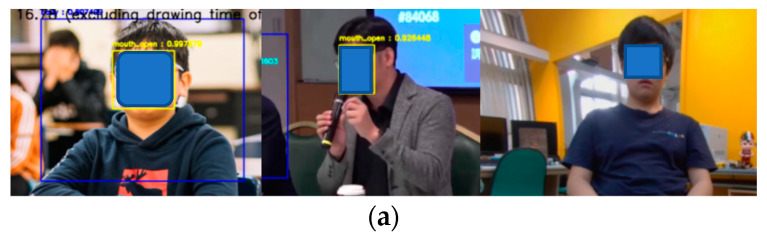
(**a**) Problems that occurred in the initial dataset. (**b**) The strong data augmentation of YOLOX.

**Figure 8 sensors-23-06250-f008:**
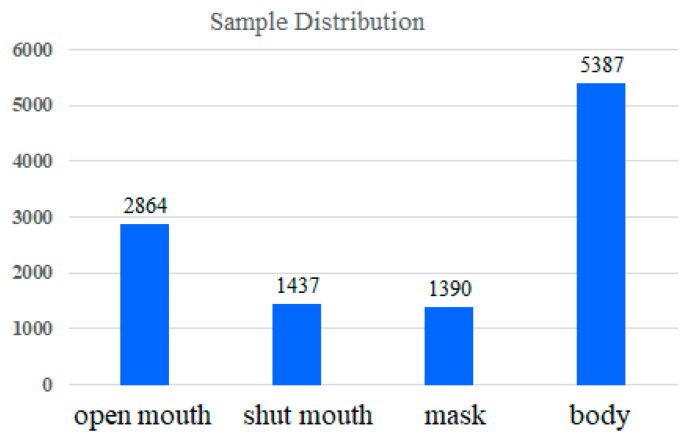
Sample distributions of the dataset.

**Figure 9 sensors-23-06250-f009:**
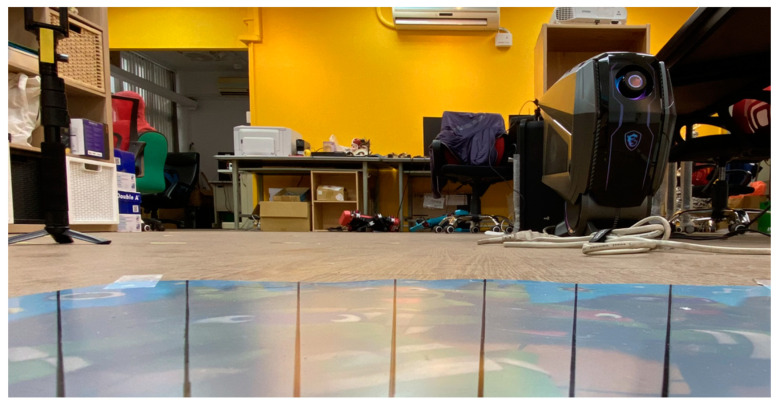
The camera view field for YOLO testing.

**Figure 10 sensors-23-06250-f010:**
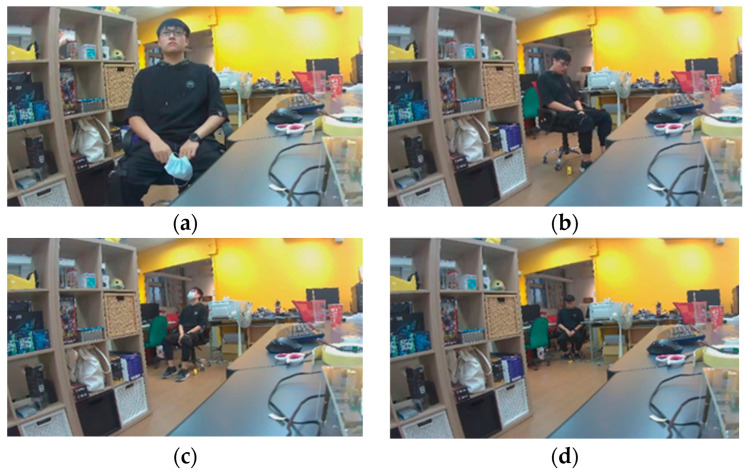
Different human states for the YOLO test. (**a**) Head-up face with the mouth closed at one meter. (**b**) Downward face with the mouth opened at two meters. (**c**) Upward face with a mask at three meters. (**d**) Downward face with the mouth opened at four meters.

**Figure 11 sensors-23-06250-f011:**
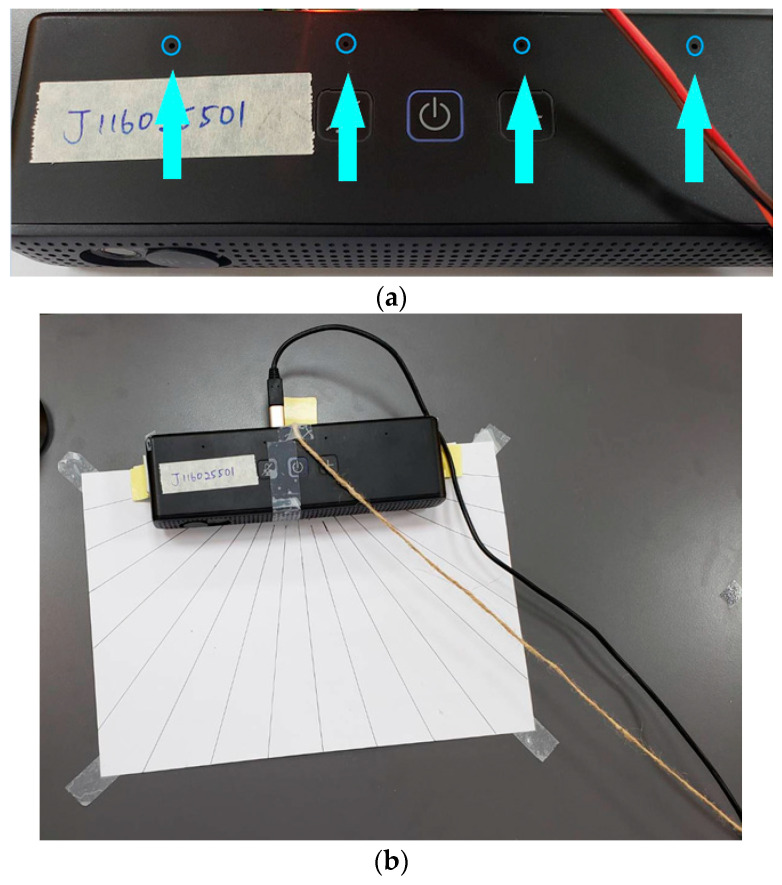
(**a**) The used microphone array. (**b**) Direction test of the TDOA.

**Figure 12 sensors-23-06250-f012:**
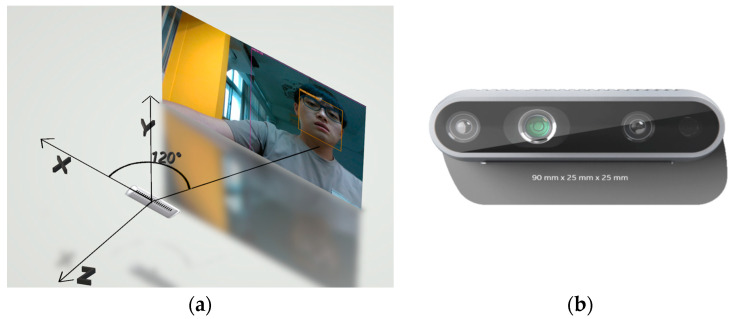
(**a**) The person’s position in the camera-centered spatial coordinates. (**b**) Intel Realsense D435 [19].

**Figure 13 sensors-23-06250-f013:**
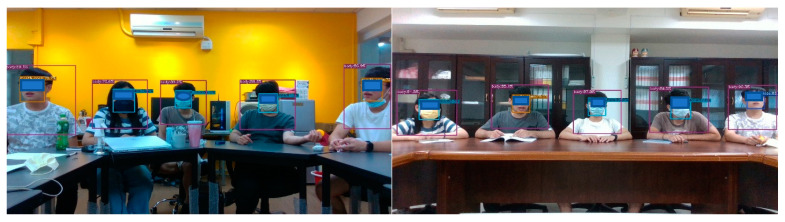
(**Left**) Space A with sufficient lighting. (**Right**) Space B with poor lighting.

**Table 1 sensors-23-06250-t001:** The performance indicators for each tested neural network on the collected dataset.

	Types	YOLOv4	YOLOv4-Tiny	Scaled YOLOv4	YOLOv5	YOLOX-Tiny	YOLOR-P6
Results							
mAP_0.5		96.92%	99.53%	94.13%	99.06%	96.21%	99.34%
Precision		85%	98%	66%	98.22%	98.52%	63.62%

**Table 2 sensors-23-06250-t002:** The recall rate of YOLOv4-tiny on the low-resolution fisheye camera. (Unit: %).

	Position (Meter)	1 M Front		1 M Left Front		1 M Right Front		2 M Front		2 M Left Front		2 M Right Front	
Mouth State													
		mouth	body	mouth	body	mouth	body	mouth	body	mouth	body	mouth	body
closed		10	100	100	98.4	12.9	0	0	0	0	0	0	0
open		88.8	90	6.5	0	100	98.4	0	0	0	0	0	0
masked		100	100	100	100	9.4	13.5	0	0	0	0	0	0
		3 M front		3 M left front		3 M right front		4 M front		4 M left front		4 M right front	
		mouth	body	mouth	body	mouth	body	mouth	body	mouth	body	mouth	body
closed		0	0	0	0	0	0	0	0	0	0	0	88.8
open		0	0	0	0	0	0	0	36.5	0	1.8	0	93
masked		0	0	0	0	0	0	0	94.7	0	70.4	0	0

**Table 3 sensors-23-06250-t003:** The recall rate of YOLOv5 on the low-resolution fisheye camera. (Unit: %).

	Position (Meter)	1 M Front		1 M Left Front		1 M Right Front		2 M Front		2 M Left Front		2 M Right Front	
Mouth State													
		mouth	body	mouth	body	mouth	body	mouth	body	mouth	body	mouth	body
closed		10	100	100	98.4	12.9	0	0	0	0	0	0	0
open		88.8	90	6.5	0	100	98.4	0	0	0	0	0	0
masked		100	100	100	100	9.4	13.5	0	0	0	0	0	0
		3 M front		3 M left front		3 M right front		4 M front		4 M left front		4 M right front	
		mouth	body	mouth	body	mouth	body	mouth	body	mouth	body	mouth	body
closed		0	97.9	0	69.8	0	24	0	0	0	0	0	0
open		0	87.1	0	75.7	0	75	0	0	0	0	0	0
masked		4.4	77.7	0	37.7	0	32.5	0	0	0	0	0	0

**Table 4 sensors-23-06250-t004:** The recall rate of YOLOX-tiny on the low-resolution fisheye camera. (Unit: %).

	Position (Meter)	1 M Front		1 M Left Front		1 M Right Front		2 M Front		2 M Left Front		2 M Right Front	
Mouth State													
		mouth	body	mouth	body	mouth	body	mouth	body	mouth	body	mouth	body
closed		100	100	75	100	64.18	100	100	100	18.3	100	14.8	100
open		88.8	100	69.1	100	100	100	0	100	97.8	100	36.6	100
masked		100	100	100	100	100	100	100	100	100	100	100	100
		3 M front		3 M left front		3 M right front		4 M front		4 M left front		4 M right front	
		mouth	body	mouth	body	mouth	body	mouth	body	mouth	body	mouth	body
closed		67.3	100	24.5	100	0	100	27.2	100	0	100	0	100
open		2.5	100	21.2	100	0	100	0	100	0	100	0	100
masked		100	100	100	100	86	100	87.7	96.5	88.5	100	97.9	100

**Table 5 sensors-23-06250-t005:** The recall rate of YOLOv4-tiny on the high-resolution non-fisheye camera. (Unit: %).

	Position (Meter)	1 M Front		1 M Left Front		1 M Right Front		2 M Front		2 M Left Front		2 M Right Front	
Mouth State													
		mouth	body	mouth	body	mouth	body	mouth	body	mouth	body	mouth	body
closed		100	100	77.1	31.4	74.1	22.2	100	100	100	100	100	100
open		100	100	100	100	87.9	81.8	100	100	100	100	100	100
masked		90	92	100	100	100	72.4	100	100	100	100	100	100
		3 M front		3 M left front		3 M right front		4 M front		4 M left front		4 M right front	
		mouth	body	mouth	body	mouth	body	mouth	body	mouth	body	mouth	body
closed		100	100	100	100	100	100	0	11.8	0	51.6	0	82.6
open		100	100	97.4	100	100	100	0	20.8	0	100	0	77.8
masked		100	100	100	100	100	100	21.1	31.6	6.9	79.3	75.9	75.9

**Table 6 sensors-23-06250-t006:** The recall rate of YOLOv5 on the high-resolution non-fisheye camera. (Unit: %).

	Position (Meter)	1 M Front		1 M Left Front		1 M Right Front		2 M Front		2 M Left Front		2 M Right Front	
Mouth State													
		mouth	body	mouth	body	mouth	body	mouth	body	mouth	body	mouth	body
closed		100	15.2	0	2.9	0	0	100	100	0	90.3	0	91.2
open		100	100	70	45	63.6	57.6	100	100	90.9	84.1	83.8	83.8
masked		62	84	0	0	0	0	73.9	97.8	5	0	3.5	10.3
		3 M front		3 M left front		3 M right front		4 M front		4 M left front		4 M right front	
		mouth	body	mouth	body	mouth	body	mouth	body	mouth	body	mouth	body
closed		0	100	0	100	0	100	0	100	80.6	100	95.7	100
open		100	100	97.4	97.4	91.7	100	8.3	100	0	100	0	100
masked		57.5	100	81.5	85.2	48.1	81.5	17.5	86	82.8	100	72.4	100

**Table 7 sensors-23-06250-t007:** The recall rate of YOLOX-tiny on the high-resolution non-fisheye camera. (Unit: %).

	Position (Meter)	1 M Front		1 M Left Front		1 M Right Front		2 M Front		2 M Left Front		2 M Right Front	
Mouth State													
		mouth	body	mouth	body	mouth	body	mouth	body	mouth	body	mouth	body
closed		100	100	94.3	100	0	96.3	100	100	100	100	100	100
open		100	100	90	100	60.6	100	100	100	97.7	100	45.9	100
masked		100	100	100	100	100	96.6	100	100	100	100	100	100
		3 M front		3 M left front		3 M right front		4 M front		4 M left front		4 M right front	
		mouth	body	mouth	body	mouth	body	mouth	body	mouth	body	mouth	body
closed		15.6	100	0	100	15.8	100	0	100	0	100	4.3	100
open		100	100	100	100	79.2	100	95.8	100	60	100	11.1	100
masked		100	100	100	100	100	100	96.5	100	100	100	100	100

**Table 8 sensors-23-06250-t008:** Confusion matrix of the TDOA positioning test results.

	Actual	30°	40°	50°	60°	70°	80°	90°
Predicted								
0°		1	0	0	0	0	0	0
27°		9	1	0	0	0	0	0
39°		0	9	0	0	0	0	0
48°		0	0	10	0	0	0	0
56°		0	0	0	8	0	0	0
64°		0	0	0	2	0	0	0
71°		0	0	0	0	10	0	0
77°		0	0	0	0	0	0	0
84°		0	0	0	0	0	10	0
90°		0	0	0	0	0	0	10
	Actual	100°	110°	120°	130°	140°	150°	
Predicted								
96°		7	0	0	0	0	0	
103°		3	3	0	0	0	0	
109°		0	7	0	0	0	0	
116°		0	0	8	0	0	0	
124°		0	0	2	8	0	0	
132°		0	0	0	2	0	0	
141°		0	0	0	0	10	2	
153°		0	0	0	0	0	8	
180°		0	0	0	0	0	0	

**Table 9 sensors-23-06250-t009:** Single person in Space A with sufficient lighting.

	Method	YOLO Only (w.o. Mask)	YOLO Only (w. Mask)	YOLO + TDOA (w.o. Mask)	YOLO + TDOA (w. Mask)
Metric					
Recall		86.3%	40.5%	82.3%	86.7%
Precision		54.8%	54.8%	100%	100%
Accuracy		78.8%	61.4%	95.6%	94.5%

**Table 10 sensors-23-06250-t010:** Single person in Space A with poor lighting.

	Method	YOLO Only (w.o. Mask)	YOLO Only (w. Mask)	YOLO + TDOA (w.o. Mask)	YOLO + TDOA (w. Mask)
Metric					
Recall		90.9%	54.0%	83.0%	65.6%
Precision		70.4%	75.4%	100%	100%
Accuracy		74.7%	54.6%	91.2%	75.4%

**Table 11 sensors-23-06250-t011:** Single person in Space B with sufficient lighting.

	Method	YOLO Only (w.o. Mask)	YOLO Only (w. Mask)	YOLO + TDOA (w.o. Mask)	YOLO + TDOA (w. Mask)
Metric					
Recall		90.5%	36.2%	67.3%	83.7%
Precision		72.4%	72.4%	100%	100%
Accuracy		78.0%	44.6%	80.4%	87.8%

**Table 12 sensors-23-06250-t012:** Single person in Space B with poor lighting.

	Method	YOLO Only (w.o. Mask)	YOLO Only (w. mask)	YOLO + TDOA (w.o. Mask)	YOLO + TDOA (w. Mask)
Metric					
Recall		53.0%	21.2%	58.5%	77.6%
Precision		72.6%	72.6%	100%	100%
Accuracy		61.7%	33.2%	76.3%	82.8%

**Table 13 sensors-23-06250-t013:** Multiple persons in Space A with sufficient lighting.

	Method	YOLO Only (w.o. Mask)	YOLO Only (w. Mask)	YOLO + TDOA (w.o. Mask)	YOLO + TDOA (w. Mask)
Metric					
Recall		64.8%	32.4%	72.0%	82.1%
Precision		27.6%	27.6%	99.7%	99.9%
Accuracy		70.7%	74.6%	96.0%	97.0%

**Table 14 sensors-23-06250-t014:** Multiple persons in Space A with poor lighting.

	Method	YOLO Only (w.o. Mask)	YOLO Only (w. Mask)	YOLO + TDOA (w.o. Mask)	YOLO + TDOA (w. Mask)
Metric					
Recall		79.8%	35.4%	71.0%	65.0%
Precision		39.0%	39.0%	100%	100%
Accuracy		78.9%	79.4%	95.8%	94.0%

**Table 15 sensors-23-06250-t015:** Multiple persons in Space B with sufficient lighting.

	Method	YOLO Only (w.o. Mask)	YOLO Only (w. Mask)	YOLO + TDOA (w.o. Mask)	YOLO + TDOA (w. Mask)
Metric					
Recall		27.3%	13.6%	43.8%	66.6%
Precision		23.7%	23.7%	100%	98.0%
Accuracy		74.3%	76.8%	91.0%	93.8%

**Table 16 sensors-23-06250-t016:** Multiple persons in Space B with poor lighting.

	Method	YOLO Only (w.o. Mask)	YOLO Only (w. Mask)	YOLO + TDOA (w.o. Mask)	YOLO + TDOA (w. Mask)
Metric					
Recall		40.6%	65.6%	40.6%	65.6%
Precision		40.4%	68.6%	98.1%	90.1%
Accuracy		82.9%	89.3%	91.4%	93.1%

## Data Availability

Not applicable.

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
