# Peer review of "Automatic Speaker Positioning in Meetings Based on YOLO and TDOA"

_sensors, 2023, doi:10.3390/s23146250_

Round 1

Reviewer 1 Report

The paper uses the objection detector YOLO to capture the upper body of all people on the screen and judge whether each person opens or closes their mouth. The experimental results show that the method proposed in this study can locate the speaker, and it has a better effect than using only one source. 

The research design is appropriate, the proposed method is adequately described and the results support the conclusions. 

It presents an interesting idea but its quality should be improved before publication. I recommend the authors to do in-depth literature review (now reference contains only 18 references) and explain the system activity diagram in detail.

Minor editing of English language required

Author Response

The paper uses the objection detector YOLO to capture the upper body of all people on the screen and judge whether each person opens or closes their mouth. The experimental results show that the method proposed in this study can locate the speaker, and it has a better effect than using only one source.

The research design is appropriate, the proposed method is adequately described and the results support the conclusions.

Response 1: Your positive comments are highly appreciated.

It presents an interesting idea but its quality should be improved before publication. I recommend the authors to do in-depth literature review (now reference contains only 18 references) and explain the system activity diagram in detail.

Response 2: Thanks for the valuable suggestion. The number of references is increased by 7 after in-depth review of some current advanced webcams. Most of them are capable to detect and track subject. However, none of them equipped with microphone array for sound source positioning.

Also, we add one more paragraph to describe the activity diagram in more detail right after it.

Reviewer 2 Report

This paper is devoted to the method of speaker extracting in the meeting room, using only the data from camera and microphone. Several existing methods of face (with/without the open mouth or with the mask) recognition have been compared to solve this task and to choose the best one. The sound recognition tehnics have also been discussed and chosen for application.

From the application point of view the paper looks interesting, and it can be considered for publication in Sensors journal after solving several issues:

1. Fig. 1: comment on the sense of YZ plane.

2.  Several figures and tables are mentioned incorrectly: Figure 2.1 (line 123), Figure 3.10 (line 422), Table 4.9 (line 583).

3. Line 430: "Equivalent" -> "equivalent".

4.  Comment on what the column names in the Table 1 mean.

5. Line 511:  "Indicates" -> "indicates".

6. Comment on what 1M-4M in Table 3 mean.

7. Line 725: better give price in dollars not in yuan.

Moderate English language editing is required.

Author Response

This paper is devoted to the method of speaker extracting in the meeting room, using only the data from camera and microphone. Several existing methods of face (with/without the open mouth or with the mask) recognition have been compared to solve this task and to choose the best one. The sound recognition technics have also been discussed and chosen for application.

From the application point of view, the paper looks interesting, and it can be considered for publication in Sensors journal after solving several issues:

  1. Fig. 1: comment on the sense of YZ plane.

Response 1: Thanks for the valuable comment. In this study, we have only horizontal microphone array which corresponds to XZ plane, so YZ plane is of no information. But, YZ plane is useful if vertical microphone array is deployed.

  1. Several figures and tables are mentioned incorrectly: Figure 2.1 (line 123), Figure 3.10 (line 422), Table 4.9 (line 583).

Response 2: Thanks for pointing out these errors.

  1. Line 430: "Equivalent" -> "equivalent".

Response 3: Corrected and thanks for the valuable comment.

  1. Comment on what the column names in the Table 1 mean.

Response 4: Thanks for the valuable comment. Table 1 is removed now.

  1. Line 511: "Indicates" -> "indicates".

Response 5: Corrected and thanks for the valuable comment.

  1. Comment on what 1M-4M in Table 3 mean.

Response 6: Thanks for the valuable comment. The unit is in meter and added in the head of Tables 2~7.

  1. Line 725: better give price in dollars not in yuan.

Response 7: Corrected and thanks for the valuable comment.

Reviewer 3 Report

This paper uses the objection detector YOLO to capture the upper body of all people on the screen and judge whether each person opens or closes their mouth. The paper is very well structured, with a detailed introduction where the contribution of this study is clearly presented, followed by a thorough literature review. The presented methodology has scientific validity and is successfully conveyed to the reader. The author provides the complete framework on the subject with an adequate volume of data, method, and well-founded and interesting results. Some minor issues should be tackled before the paper is accepted for publication: 

1. when the author calls the table, figure, or equation, it should have the hyperlink to recognize where it is easily. The template of the MDPI manuscript has this one.

2. As shown in Table 7, the data distribution is unequal. How can the author solve this obstacle?

Author Response

This paper uses the objection detector YOLO to capture the upper body of all people on the screen and judge whether each person opens or closes their mouth. The paper is very well structured, with a detailed introduction where the contribution of this study is clearly presented, followed by a thorough literature review. The presented methodology has scientific validity and is successfully conveyed to the reader. The author provides the complete framework on the subject with an adequate volume of data, method, and well-founded and interesting results. Some minor issues should be tackled before the paper is accepted for publication:

  1. when the author calls the table, figure, or equation, it should have the hyperlink to recognize where it is easily. The template of the MDPI manuscript has this one.

Response 1: Hyperlink added for each table, figure, and equation. Thanks for the valuable comments.

  1. As shown in Figure 7, the data distribution is unequal. How can the author solve this obstacle?

Response 2: Thanks for the valuable comments. In this study, the data distribution is not balanced and data augmentation is often used to increase the dataset size. Hopefully, the gaps among the categories are not too large and simply resize, rotation, and flip could make it.